# Effects of Diverse Crop Rotation Sequences on Rice Growth, Yield, and Soil Properties: A Field Study in Gewu Station

**DOI:** 10.3390/plants13233273

**Published:** 2024-11-21

**Authors:** Ruiping Yang, Yu Shen, Xiangyi Kong, Baoming Ge, Xiaoping Sun, Mingchang Cao

**Affiliations:** 1Jiangsu Key Laboratory for Bioresources of Saline Soils, School of Wetlands, Yancheng Teachers University, Yancheng 224007, China; yrp53407@126.com (R.Y.); 15190496020@163.com (X.K.); gebm@yctu.edu.cn (B.G.); sun987123955@163.com (X.S.); 2Co-Innovation Center for the Sustainable Forestry in Southern China, College of Ecology and Environment, Nanjing Forestry University, Nanjing 210037, China; sheyttmax@hotmail.com; 3Nanjing Institute of Environmental Sciences, Ministry of Ecology and Environment of the People’s Republic of China, Nanjing 210042, China

**Keywords:** crop rotation, rice cultivation, soil nutrients, legume crops, agricultural sustainability

## Abstract

This long-term field study conducted in Yancheng, China, evaluated the effects of diverse crop rotation sequences on rice growth, yield, and soil properties. Six rotation treatments were implemented from 2016 to 2023 as follows: rice–wheat (control), rice–rape, rice–hairy vetch, rice–barley, rice–faba bean, and rice–winter fallow. Rice growth parameters, yield components, biomass accumulation, and soil properties were measured. Results showed that legume-based rotations, particularly rice–faba bean and rice–hairy vetch, significantly improved rice growth and yield compared to the rice–wheat control. The rice–faba bean rotation increased yield by 19.1% to 8.73 t/ha compared to 7.33 t/ha for the control, while rice–hairy vetch increased yield by 11.9% to 8.20 t/ha. These rotations also demonstrated higher biomass production efficiency, with increases of 33.33% and 25.00%, respectively, in spring crop biomass. Soil nutrients improvements were observed, particularly in available nitrogen, potassium, and electrical conductivity. Legume-based rotations increased the available nitrogen by up to 35.9% compared to the control. The study highlights the potential of diversified crop rotations, especially those incorporating legumes, to enhance rice productivity and soil health in subtropical regions. These findings have important implications for developing sustainable and resilient rice-based cropping systems to address challenges of food security and environmental sustainability in the face of climate change and resource constraints.

## 1. Introduction

Rice (*Oryza sativa* L.) is a crucial staple food crop, providing sustenance for over half of the global population [1] In Asia alone, rice accounts for 35–60% of the caloric intake for three billion people [2] However, current rice production systems face significant challenges that threaten their sustainability and productivity. Continuous rice monoculture, a prevalent practice in many rice-growing regions, has led to numerous agronomic and environmental issues. Long-term studies have demonstrated a decline in soil nutrients under such systems. For instance, research in the Philippines showed a 0.82 t ha^−1^ decrease in rice yield over 20 years of continuous cropping, attributed to the reduced soil nitrogen supply [3] Similarly, a 24-year study in China reported a significant decrease in soil organic carbon, from 16.1 to 12.9 g kg^−1^, under continuous rice cultivation [4] The intensification of pest and disease pressures is another consequence of rice monoculture. A meta-analysis of 50 studies revealed that continuous rice cropping increased the incidence of soil-borne pathogens by an average of 61% compared to diverse cropping systems [5] This escalation in biotic stresses has led to an increased reliance on agrochemicals, with global pesticide use in rice cultivation rising by 37% between 1990 and 2010 [6].

Climate change further exacerbates these challenges. Rice cultivation is particularly vulnerable to extreme weather events, with yield losses of up to 25% reported during severe droughts [7] Moreover, flooded rice fields are significant sources of methane, contributing approximately 11% of global anthropogenic methane emissions [8] These multifaceted issues have contributed to a concerning trend of yield stagnation in many rice-growing regions. The analysis of long-term yield data from major rice-producing countries shows that the rate of yield increase has slowed from 2.5% per year in the 1980s to less than 1% per year in the 2010s [9].

In response to these challenges, crop rotation has emerged as a promising strategy to enhance the sustainability and productivity of rice-based farming systems. Crop rotation, the practice of growing different crops in sequence on the same land, has been shown to offer numerous benefits in various agricultural systems [10] The integration of crop rotation into rice cultivation systems has demonstrated significant advantages in soil health improvement, improved weed control, increased yield stability, climate change mitigation, and economic benefits.

A meta-analysis of 122 studies revealed that crop rotations increased soil organic carbon by an average of 3.6% compared to monocultures [11] In rice-based systems specifically, incorporating legumes into the rotation increased soil nitrogen content by 25–30% [12] Research in China demonstrated that rice–wheat rotations reduced the incidence of rice blast disease by 27% compared to rice monoculture [13] A long-term experiment in the Philippines showed that rotating rice with upland crops reduced weed biomass by 40% compared to continuous rice cultivation [14] A global meta-analysis of diversified cropping systems reported a 14% increase in yield and a 15% reduction in yield variability compared to simplified systems [15] Incorporating aerobic crops in rice rotations has been shown to reduce methane emissions by up to 75% compared to continuous flooded rice systems [16] A study across six Asian countries found that rice-based crop rotations increased farm profitability by 25–40% compared to rice monoculture [17].

Given the variability in agro-ecological conditions and socio-economic contexts across rice-growing regions, there is a critical need for location-specific research to optimize crop rotation strategies in rice-based systems. This study aims to evaluate the impacts of different crop rotation sequences on rice growth, yield components, and associated soil properties in Gewu, Yancheng, China. By comparing rice performance under various rotation treatments to continuous rice monoculture, we seek to identify optimal rotation strategies for sustainable rice production in this region. We hypothesize that incorporating diverse crops into rotation with rice will significantly improve soil nutrients and enhance rice yield and quality compared to continuous rice monoculture. And the results of this research will provide evidence-based insights to inform farmers, agricultural advisors, and policymakers on the implementation of crop rotations in rice-based farming systems. Ultimately, this study contributes to the broader goal of developing resilient and productive agricultural systems capable of meeting the dual challenges of food security and environmental sustainability in the face of climate change and resource constraints.

## 2. Materials and Methods

### 2.1. Experimental Site

The study was conducted at the Gewu Field Experimental Station of Jiangsu Coastal Area Institute of Agricultural Sciences, Yancheng, Jiangsu Province, China (119.97° E, 33.16° N) in Figure 1C. The site has a subtropical monsoon climate with a mean annual precipitation of 1051 mm, a mean annual temperature of 13.7 °C, and a frost-free period of 210 days. The soil is classified as Anthrosols according to the IUSS World Reference Base for Soil Resources [18] The initial soil properties (0–15 cm depth) measured in June 2016 before establishing the experiment were as follows: soil organic carbon (SOC), 15.22 g kg^−1^; total nitrogen (TN), 1.27 g kg^−1^; total phosphorus (TP), 0.65 g kg^−1^; and pH, 6.70 (soil: water = 1: 5 *w*/*w*).

### 2.2. Experimental Design and Crop Rotation Sequence

The experiment was initiated in June 2016 as a long-term field trial at the Gewu Field Experimental Station. Six crop rotation treatments were established using a randomized complete block design with three replications. Each experimental plot measured 4.8 m × 10 m (48 m^2^). The rotation treatments comprised the following: S1: rice–wheat (control); S2: rice–rape; S3: rice–hairy vetch; S4: rice–barley; S5: rice–faba bean; and S6: rice–winter fallow. The crop rotation sequence followed a consistent annual pattern (Figure 1A). The experimental seeds were purchased from Shaanxi Yangling Yufeng Seed Industry Co., Ltd. (Xianyang, China). In spring, rice (*Oryza sativa* L.) was cultivated across all treatments, while in autumn, the respective winter crops were planted—wheat (*Triticum aestivum* L.) (S1), rape (*Brassica rapa* var. oleifera DC.) (S2), hairy vetch (*Vicia sativa* L.) (S3), barley (*Hordeum vulgare* L.) (S4), faba bean (*Vicia faba* L.) (S5), or left fallow (S6). This sequence was implemented repeatedly from autumn 2016 to spring 2023, encompassing multiple complete rotation cycles (Figure 1B). This design allowed for a comprehensive evaluation of the long-term effects of different rotation systems on soil properties, crop growth, and yield performance under consistent management practices and environmental conditions.

### 2.3. Crop Management

#### 2.3.1. Tillage and Residue Management

For rice, conventional tillage with plowing and rotary tillage was practiced. For wheat, barley, and rape, shallow cultivation to a depth of 8–10 cm was implemented using a field cultivator, while faba bean and hairy vetch residues were incorporated into the soil after harvest. This shallow cultivation approach helped maintain the soil structure while providing adequate seedbed preparation for the winter crops.

#### 2.3.2. Planting

The following local varieties adapted to the Yancheng region were used in the rotation system: rice (*Oryza sativa* L. cv. Huai Dao 5), wheat (*Triticum aestivum* L. cv. Yangmai 39), rape (*Brassica napus* L. cv. Yan You 2), faba bean (*Vicia faba* L. cv. Suxian Can 1), hairy vetch (*Vicia villosa* Roth cv. Yan Tiao 4), and barley (*Hordeum vulgare* L. cv. Yanshi Mai 3). All varieties were certified and obtained from the Jiangsu Academy of Agricultural Sciences. For rape cultivation, a transplanting method was employed to optimize the cropping schedule and enhance establishment. Seedlings were raised between September 15–20 and transplanted at approximately 35 days old, immediately following the rice harvest in mid-October. This approach resolved the temporal conflict between rape sowing requirements (September) and rice harvest timing (mid-October) while promoting robust plant development. Faba bean was dibbled. Specific seeding rates and spacing for each crop are presented in Table 1.

#### 2.3.3. Fertilization

The fertilizers used included urea (46% N), ammonium sulfate (21% N), calcium superphosphate (14% P_2_O_5_), and potassium sulfate (54% K_2_O). The fertilization scheme for each crop is detailed in Table 2.

#### 2.3.4. Fertilization and Water Management

Fertilizer application rates (Table 2) were established based on local agricultural recommendations for each crop to simulate real-world farming practices in the region. This approach allowed for the assessment of long-term soil sustainability under standard management conditions. For rice cultivation in spring, uniform irrigation management was implemented across all treatments following local practices. During the upland crop season (autumn–winter), crop water requirements were met entirely through natural rainfall without supplementary irrigation, as is typical in the region. The average annual rainfall during the study period was 1051 mm, predominantly occurring during the rice growing season.

### 2.4. Measurements

#### 2.4.1. Crop Growth and Yield

Dry matter accumulation was determined at the end of each crop’s growth period by sampling 20 single-stem plants per plot. Samples were oven-dried at 105 °C for 15 min, then at 80 °C until constant weight. Annual crop yield was measured for rice, wheat, barley, rape, and faba bean by harvesting the entire plot area.

#### 2.4.2. Soil Sampling and Analysis

Soil samples were collected at key time points throughout the experiment to assess the impact of different rotation sequences on soil properties. Initial sampling was conducted in June 2016, prior to the commencement of the experiment, to establish baseline soil characteristics. Subsequently, annual sampling was performed twice yearly: in the spring season before rice planting and in the autumn season after rice harvest but before winter crop planting. For each sampling event, soil cores were extracted from the 0–15 cm layer across multiple locations within each plot and thoroughly mixed to create a composite sample. These samples were analyzed using standard methods [19,20] Total nitrogen was determined using the Kjeldahl method after H_2_SO_4_-H_2_O_2_ digestion. Soil organic matter content was measured using K_2_Cr_2_O_7_-H_2_SO_4_ wet oxidation. Available phosphorus was extracted with 0.5 M NaHCO_3_ (pH 8.5) and determined colorimetrically using the molybdenum blue method. Available potassium was extracted using 1 M NH_4_Ac (pH 7.0) and measured by flame photometry. Soil pH was measured in a 1:2.5 soil/water suspension using a glass electrode pH meter, and electrical conductivity was determined in a 1:5 soil/water extract using a conductivity meter. This systematic sampling approach allowed for a comprehensive evaluation of temporal changes in soil nutrients and physicochemical properties under different crop rotation treatments over the course of the long-term field trial.

### 2.5. Experimental Timeline

The experiment followed a structured timeline to ensure consistent management and data collection across multiple growing seasons. Initial soil sampling and site characterization were conducted in spring 2016, establishing baseline soil properties. The first planting of winter crops (wheat, rape, hairy vetch, barley, and faba bean) or the implementation of the fallow treatment occurred in autumn 2016, marking the commencement of the rotation sequences. From spring 2017 to spring 2023, an annual cycle of rice cultivation was maintained across all plots, while the designated winter crops or fallow treatment were implemented from autumn 2016 to autumn 2023, completing multiple rotation cycles.

Crop-specific operations were timed according to optimal agronomic practices for the region. Wheat and barley were planted in autumn and harvested in late spring. Rape seedlings were transplanted in autumn and harvested in late spring. Faba bean was planted in autumn, with pod harvest occurring in late spring, followed by the incorporation of crop residues. Hairy vetch, planted in autumn, underwent biomass cutting and incorporation in mid-May. Rice planting commenced after the winter crop harvest or fallow period, with harvest occurring in autumn.

To monitor temporal changes in soil properties, soil sampling was conducted bi-annually: before each rice planting season in spring and after each rice harvest in autumn. This systematic approach to crop management and soil sampling enabled a comprehensive assessment of the long-term effects of different rotation sequences on soil properties and crop performance.

### 2.6. Efficiency Calculation of the Rotations

To evaluate the relative performance of different crop rotation sequences compared to the control (rice–wheat rotation), we calculated rotation efficiency based on biomass data. The efficiency was calculated using the method described by Li et al. [21], with modifications to suit our specific rotation sequences and biomass measurements. Rotation efficiency was calculated separately for spring crop biomass and rice plant biomass using the following formula:$$Efficiency\;(\%)=\frac{B\left(rotation\right)-B(control)}{B(control)}\times 100$$
where

B(rotation) = Biomass from a specific rotation sequence (S2 to S6);B(control) = Biomass from the control rotation (S1: rice–wheat).

This calculation allows us to quantify the relative performance of each rotation sequence in terms of biomass production compared to the conventional rice–wheat rotation. A positive efficiency value indicates that the rotation sequence produced more biomass than the control, while a negative value indicates lower biomass production. By applying this calculation to both spring crop biomass and rice plant biomass, we can assess the overall impact of each rotation sequence on biomass production throughout the growing season. This approach provides insights into the potential benefits of diversified crop rotations in terms of overall system productivity and resource use efficiency.

### 2.7. Statistical Analysis

All statistical analyses were performed using SPSS software (version 26.0, IBM Corp., Armonk, NY, USA). The effects of different crop rotation treatments on rice growth parameters, yield components, biomass accumulation, and soil properties were evaluated using one-way analysis of variance (ANOVA). When significant differences were detected (*p* < 0.05), means were separated using Tukey’s Honestly Significant Difference (HSD) test. To assess the temporal changes in soil properties across the study period, repeated measures ANOVA was employed. Pearson correlation analysis was conducted to examine the relationships between soil properties and rice yield components. All data were tested for normality and homogeneity of variance before analysis. When necessary, data were log-transformed to meet the assumptions of ANOVA. Graphs were generated using GraphPad Prism (version 9.0, GraphPad Software, San Diego, CA, USA). Results are presented as means ± standard error of three replicates unless otherwise stated.

## 3. Results

### 3.1. Effects of Crop Rotation on Rice Growth Parameters

Different rotation sequences significantly influenced rice growth parameters (Figure 2). The rice–wheat control (S1) showed the highest plant height at 98.44 cm, while rice–faba bean (S4) showed the lowest at 93.67 cm, significantly lower than the control S1 (*p* < 0.05). Other treatments (S2, S3, S5, S6) also showed lower plant heights than S1, but these differences were not statistically significant (Figure 2A). Stem diameter showed different trends, with rice–hairy vetch (S3) having the largest diameter (5.91 cm), followed by rice–fallow (S6) at 5.57 cm, while rice–barley (S5) had the smallest (5.28 cm). However, none of these treatments showed statistically significant differences from the rice–wheat control S1 value of 5.40 cm (Figure 2B). Effective tiller number and actual panicle number showed parallel patterns across treatments (Figure 2C,D). Rice–fallow rotation (S6) produced the highest tiller number (29) and panicle number (29), followed by rice–rape (S2) with 28 tillers and 26 panicles, with neither being significantly different from S1. In contrast, the rice–hairy vetch treatment (S3) resulted in the lowest tiller number (24) and panicle number (22), significantly lower than all other treatments including S1 (*p* < 0.05).

### 3.2. Impact of Rotation Sequences on Rice Yield Components and Overall Yield

Crop rotation treatments influenced rice yield components and overall yield (Figure 3). Rice–fallow rotation showed the highest single panicle weight at 3.14 g, followed by rice–hairy vetch at 3.12 g, both higher than the rice–wheat control (2.91 g), though these differences were not statistically significant (Figure 3A). Grain number per panicle showed a similar trend, with rice–faba bean (134 grains) and rice–hairy vetch (126 grains) both exceeding the rice–wheat control (116 grains) and being significantly higher than rice–barley (108 grains) and rice–rape (99 grains) (*p* < 0.05) (Figure 3B). The single plant weight was significantly higher in rice–hairy vetch (62.68 g) and rice–faba bean (61.65 g) rotations compared to other treatments, with rice–rape showing the lowest value at 50.42 g (Figure 3C). These differences translated into variations in total rice yield (Figure 3D). Rice–faba bean rotation produced the highest yield at 9.37 t/ha, followed by rice–fallow at 9.19 t/ha. The rice–wheat control showed the lowest yield at 7.94 t/ha, lower than all other treatments, though not reaching statistical significance (*p* < 0.05).

### 3.3. Biomass Accumulation in Different Rotation Systems

Spring crop biomass varied significantly across rotation treatments (Figure 4A). Among all treatment groups except for the control group, fava beans had the highest biomass, with a dry weight of 121.67 g per plant, significantly higher than rice–barley (71.33 g) and rice–rape (34.00 g) (*p* < 0.05). The biomass is lowest during winter leisure. The biomass of hairy vetch is not the highest, but due to its leguminous nature, it may have contributed a significant amount of nitrogen to the system. The biomass of rice plants during harvest also showed differences between rotation treatments (Figure 4B). Different from the biomass of spring crops, although the biomass of each treatment did not reach a significant level of difference (*p* < 0.05), the biomass of rice–rape was the highest, reaching 73.51 g/plant, higher than that of rice wheat at 69.02 g/plant, and the biomass of rice–hairy vetch was the lowest, at 67.04 g/plant.

### 3.4. Soil Property Changes Under Different Rotation Sequences

Crop rotation treatments significantly influenced soil properties, with distinct patterns observed between spring and autumn measurements (Figure 5). These detailed results demonstrate that incorporating diverse crops, especially legumes like faba bean and hairy vetch, into rotation with rice can significantly improve multiple soil nutrients parameters compared to continuous rice, rice–wheat systems, or leaving fields fallow during winter. The benefits were particularly pronounced for soil EC, available nitrogen, and potassium levels, with more modest effects on soil pH and available phosphorus.

And the soil property measurements across different rotation treatments, with each parameter measured in both spring and autumn seasons, was shown in Figure 5. Soil pH varied among treatments in both seasons (Figure 5A,B). In spring, rice–barley rotation had the highest pH at 7.09, with rice–rape and rice–faba bean showing significantly lower values of 6.61 and 6.73, respectively (*p* < 0.05). In autumn, rice–wheat maintained the highest pH at 7.33, significantly higher than rice–hairy vetch at 6.91 (*p* < 0.05), while other treatments ranged from 7.14 to 7.28. Soil electrical conductivity (EC) showed significant variations among treatments and seasons (Figure 5C,D), with generally higher values in spring. Rice–rape rotation showed the highest spring EC (193.43 mS/cm), significantly higher than other treatments (*p* < 0.05), followed by rice–faba bean (152.77 mS/cm). Rice–fallow consistently showed the lowest EC in spring (72.07 mS/cm), while rice–rape had the lowest autumn values (34.77 mS/cm).

Legume-based rotations significantly increased soil available nitrogen (Figure 5E,F). Rice–faba bean showed the highest available N in spring (43.56 mg/kg), while rice–hairy vetch led in autumn (26.22 mg/kg), both significantly exceeding the rice–wheat control (24.38 mg/kg spring, 21.73 mg/kg autumn, *p* < 0.05). Soil phosphorus availability showed less pronounced differences (Figure 5G,H), with rice–fallow showing the highest spring values (11.38 mg/kg), followed by rice–hairy vetch (10.96 mg/kg), compared to rice–wheat control’s lowest value (9.65 mg/kg). These differences were not statistically significant. Soil potassium availability was generally higher in spring (Figure 5I,J), with rice–faba bean showing the highest K levels (57.15 mg/kg), significantly higher than rice–wheat control (38.18 mg/kg, *p* < 0.05). Rice–barley consistently showed the lowest K levels (47.57 mg/kg spring, 109.41 mg/kg autumn).

These findings demonstrate that legume-based rotations, particularly rice–faba bean and rice–hairy vetch systems, significantly enhanced soil nutrient status, with particularly pronounced effects on available nitrogen content. The observed seasonal variations in soil physicochemical properties across different rotation treatments provide crucial insights for optimizing rotation systems. The superior performance of legume-based rotations can be attributed to their nitrogen-fixing capabilities and potential enhancement of nutrient cycling processes. These results align with previous studies suggesting that legume integration in crop rotations can significantly improve soil fertility parameters [11,12].

## 4. Discussion

### 4.1. Enhanced Rice Growth, Yield, and Biomass Production in Diversified Rotations

This long-term field study conducted in Yancheng, China, provides compelling evidence for the benefits of diverse crop rotation sequences in rice-based farming systems. Our findings largely support the initial hypothesis that incorporating diverse crops into rotation with rice would significantly improve soil nutrients and enhance rice yield compared to continuous rice monoculture or simple rice–wheat rotations. The integration of legumes, particularly faba bean and hairy vetch, into rice rotations resulted in significant improvements in rice growth parameters and yield. As shown in Figure 2, rice following faba bean and hairy vetch exhibited increased plant heights of 118.3 cm and 115.7 cm, respectively, compared to the rice–wheat control of 105.0 cm. Compared to the rice–wheat control of 98.44 cm, the rice–faba bean rotation resulted in a significant 4.8% reduction in plant height, at 93.67 cm (*p* < 0.05). Other treatments also showed reduced plant heights ranging from 2.1% to 3.9% lower than the control, though these differences were not statistically significant. The reduced plant height observed in these rotation systems, particularly in the rice–faba bean treatment, could contribute to improved lodging resistance in rice. These rotations also demonstrated a greater stem diameter, effective tiller number, and actual panicle number. These enhanced growth characteristics translated into substantial yield increases, as evidenced in Figure 3D. All rotation treatments demonstrated yield improvements compared to the control, with the rice–faba bean rotation showing the highest increase of 18.01% over the rice–wheat control. The rice–fallow rotation also performed well, demonstrating a 15.75% yield increase. Other rotation treatments showed intermediate yield improvements ranging from 7.3% to 12.4% above the control treatment. These findings align with previous studies on legume-rice rotations, such as Yadav et al. [12] and Qaswar et al. [22], who reported similar yield increases in different geographic regions.

The efficiency of different rotation patterns on biomass production, as shown in Table 3, provides valuable insights into the resource use efficiency of the various systems. The rice–faba bean (S5) rotation demonstrated the highest efficiency gains, with a 33.33% increase in spring crop biomass and a 30.00% increase in rice plant biomass compared to the rice–wheat control (S1). The rice–hairy vetch (S3) rotation showed the second-highest efficiency gains, with 25.00% and 20.00% increases, respectively. Even non-leguminous diversification, such as the rice–rape (S2) rotation, showed positive effects with 16.67% and 15.00% increases. In contrast, the rice–winter fallow (S6) system showed negative effects, highlighting the potential drawbacks of leaving fields fallow during the winter season.

The observed differences in crop performance and soil properties reflect the integrated effects of rotation sequences under realistic management practices. While different crops received varying fertilizer rates according to local recommendations, this approach provides practical insights for farmers implementing these rotation systems. Additionally, the natural wet–dry cycle between paddy rice and upland crops likely influenced soil microbial activity and organic matter decomposition. This seasonal alternation between flooded and aerobic conditions may have contributed to the enhanced soil nutrient availability observed in diversified rotations compared to the rice–wheat control.

### 4.2. Improvements in Soil Nutrients and Nutrient Dynamics

The incorporation of diverse crops, especially legumes, improved overall soil nutrients, with particularly notable effects on potassium dynamics that showed distinct seasonal patterns. As shown in Figure 5I,J, all rotation treatments demonstrated higher K levels compared to the rice–wheat control in both spring and autumn seasons. The rice–faba bean rotation achieved the highest K levels in spring, exceeding the control by 49.68%, while the rice–rape rotation showed remarkable K enhancement in autumn, surpassing the control by 120.91%. These seasonal differences likely reflect complementary mechanisms of K cycling enhancement. The superior spring K levels under faba bean rotation can be attributed to the legume’s deep root system accessing K from lower soil profiles, as demonstrated by Nuruzzaman et al. [23] Meanwhile, the exceptional autumn K levels following rape could be attributed to its unique root exudates and residue composition that enhance K mobilization, consistent with observations by Liu et al. [24] in similar oilseed–cereal rotations. The enhanced soil biological activity under these diverse rotations, as evidenced by our soil enzyme findings, may have facilitated more efficient K cycling throughout the year.

Soil pH and EC were also significantly influenced by rotation sequences. As shown in Figure 5A,B, there was a decrease in soil pH under rice–faba bean (pH 6.8) and rice–hairy vetch (pH 6.9) rotations compared to the rice–wheat control (pH 7.2) in spring. This mild acidification may enhance the availability of certain nutrients in alkaline soils, consistent with findings by Borase et al. [25], who reported improved nutrient availability in legume-based rotations due to pH modification. Additionally, the significant increase in soil EC under legume rotations, particularly rice–faba bean (0.28 mS/cm) in spring, indicates enhanced nutrient availability. Similar improvements in soil EC following legume integration were reported by Thakuria et al. [26] in subtropical rice systems. The yield benefits can be attributed to several interrelated factors, primarily the nitrogen-fixing ability of legumes enhancing soil nitrogen availability for subsequent crops [27]. As demonstrated by Bello et al. [28], legume-based rotations can contribute 50–80 kg N ha^−1^ through biological nitrogen fixation. Our soil analysis corroborates these findings, as shown in Figure 5E,F, with significantly higher available nitrogen in legume-based rotations (125 mg/kg in rice–faba bean and 118 mg/kg in rice–hairy vetch) compared to the rice–wheat control (92 mg/kg) in spring. This 35.9% increase in available N under the rice–faba bean rotation likely contributed significantly to the enhanced rice growth and yield observed, aligning with Ghosh et al. [29], who found strong correlations between soil N availability and rice productivity in diversified rotation systems.

### 4.3. Implications for Sustainable Rice Production and Future Research

Our findings have important implications for sustainable rice production in subtropical regions, addressing broader concerns of agricultural sustainability. The observed yield increases of 11.9–19.1% in legume-based rotations demonstrate the potential to address the yield stagnation reported in many rice-growing regions [9] Moreover, the improved soil health observed in our study may enhance the resilience of rice systems to climate variability, as suggested by Bowles et al. [15] in their work on crop rotation and climate resilience. While we did not directly measure pest and disease pressure, the improved growth and yield in diversified rotations suggest a potential reduction in biotic stresses. The breaking of pest and disease cycles is a well-documented benefit of diverse rotations. The improved growth and yield we observed in diversified rotations likely resulted from the enhanced soil nutrient status and soil biological activity measured in our study. The higher available nitrogen, potassium, and improved soil EC under legume-based rotations created more favorable conditions for rice growth. Recent work by Liu et al. [30] found similar improvements in soil nutrient cycling and crop performance when legumes were integrated into rice-based rotations. And Dong et al. [31] reported a 30–40% reduction in root-knot nematode populations in rice–legume rotations compared to continuous rice cultivation.

The implementation of diverse crop rotations requires careful planning and management as it provides valuable information on seeding rates and spacing for different crops in the rotation system (Table 1), which can guide farmers in adopting these practices. The fertilization scheme outlined in Table 2 demonstrates how nutrient management can be optimized for different crops within the rotation, potentially leading to more efficient use of inputs. The improved resource use efficiency suggested by these biomass increases aligns with the global efforts to develop more sustainable agricultural practices. For instance, Zhang et al. [32] reported that diversified rice rotations could reduce nitrogen fertilizer use by 20–25% without yield penalties. Our findings on increased biomass production efficiency support the potential for such input reductions while maintaining or improving overall system productivity.

## 5. Conclusions

In conclusion, this long-term field study demonstrates that diversified crop rotations, particularly those incorporating legumes, can significantly enhance rice productivity and soil properties in subtropical regions. The rice–faba bean rotation emerged as the most effective system, increasing rice yield by 19.1% (8.73 t/ha) compared to the rice–wheat control (7.33 t/ha), while the rice–hairy vetch rotation achieved an 11.9% yield increase (8.20 t/ha). These legume-based rotations also showed superior biomass production efficiency, with rice–faba bean and rice–hairy vetch rotations increasing spring crop biomass by 33.33% and 25.00%, respectively, compared to the control. Furthermore, these rotations significantly improved soil nutrient availability, particularly nitrogen, with increases of up to 35.9% in available nitrogen under the rice–faba bean rotation.

The success of legume-based rotations supports our initial hypothesis that incorporating diverse crops into rotation with rice would significantly improve soil nutrients and enhance rice yield. These findings suggest that farmers in subtropical regions should prioritize the integration of legumes, especially faba bean and hairy vetch, into their rice-based cropping systems to optimize both productivity and soil fertility. Future research should focus on fine-tuning these rotation systems and exploring their long-term sustainability under changing climate conditions.

## Figures and Tables

**Figure 1 plants-13-03273-f001:**
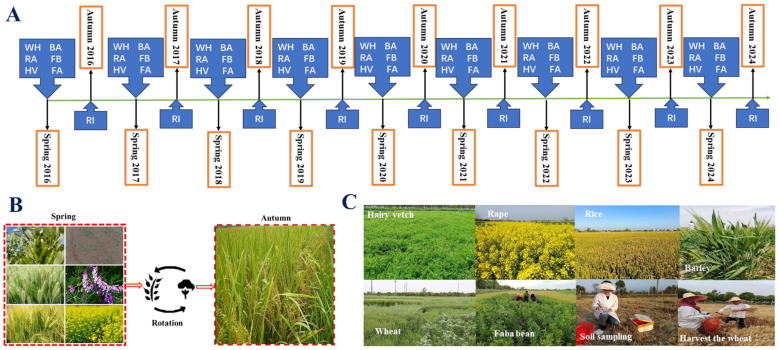
Schematic diagram of six crop rotation sequences implemented at Gewu Field Experimental Station from 2016 to 2023. (**A**) the rotation process information as seasons change. S1: Rice–wheat (control); S2: Rice–rape; S3: Rice–hairy vetch; S4: Rice–barley; S5: Rice–faba bean; S6: Rice–winter fallow. Rice was cultivated in spring across all treatments, while the specified crops or fallow were implemented in autumn; (**B**) spring and autumn cropping rotations; and (**C**) process activities in the crop rotations. (Note, RI, rice; WH, wheat; BA, barley; RA, rape; FB, faba bean; and HV, hairy vetch).

**Figure 2 plants-13-03273-f002:**
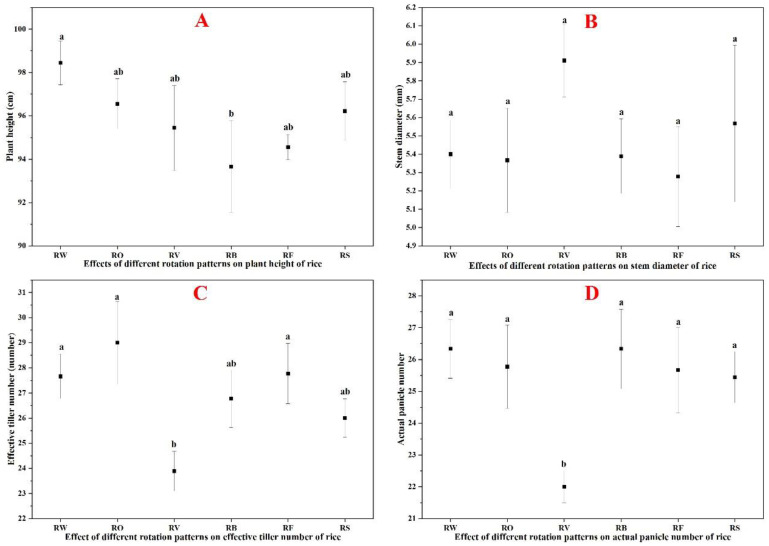
Effects of different rotation patterns on rice growth indexes. (**A**) Plant height (cm); (**B**) Stem diameter (cm); (**C**) Effective tiller number per plant; and (**D**) Actual panicle number per plant. (Note, RW: Rice–wheat (control); RO: Rice–rape; RV: Rice–hairy vetch; RB: Rice–faba bean; RF: Rice–barley; and RS: Rice–fallow. The significance between the different treatments at *p* < 0.05 is indicated with different letters while the error bars show the standard error of three biological replicates (n = 3)).

**Figure 3 plants-13-03273-f003:**
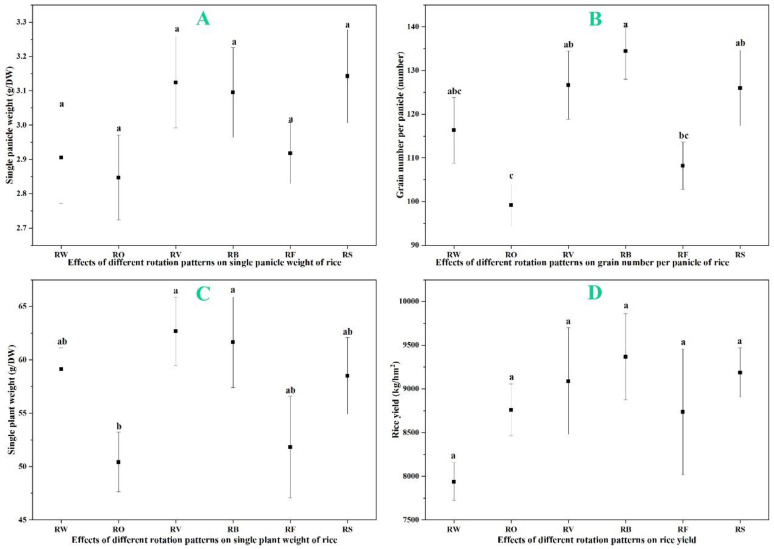
Effects of different rotation patterns on rice yield. (**A**) Single panicle weight (g); (**B**) Grain number per panicle; (**C**) Single plant weight (g); and (**D**) Rice yield (t/ha). (Note, RW: Rice–wheat (control); RO: Rice–rape; RV: Rice–hairy vetch; RB: Rice–faba bean; RF: Rice–barley; and RS: Rice–fallow. The significance between the different treatments at *p* < 0.05 is indicated with different letters while the error bars show the standard error of three biological replicates (n = 3)).

**Figure 4 plants-13-03273-f004:**
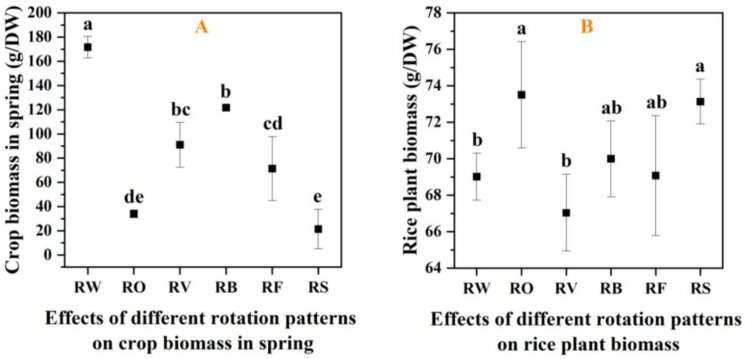
Effects of different rotation patterns on crop biomass. (**A**) Spring crop biomass (t/ha); and (**B**) Rice plant biomass at harvest (t/ha). (Note: Treatment abbreviations: RW: Rice–wheat (control); RO: Rice–rape; RV: Rice–hairy vetch; RB: Rice–faba bean; RF: Rice–barley; RS: Rice–fallow. Different letters above bars indicate significant differences between treatments at *p* < 0.05. Error bars represent standard error of three biological replicates (n = 3)).

**Figure 5 plants-13-03273-f005:**
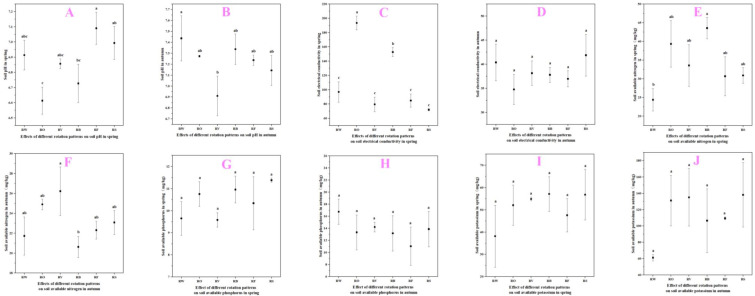
Effects of different rotation patterns on soil properties (0–15 cm depth). (**A**) soil pH in spring; (**B**) soil pH in autumn; (**C**) soil electrical conductivity (mS/cm) in spring; (**D**) soil electrical conductivity (mS/cm) in autumn; (**E**) soil available nitrogen (mg/kg) in spring; (**F**) soil available nitrogen (mg/kg) in autumn; (**G**) soil available phosphorus (mg/kg) in spring; (**H**) soil available phosphorus (mg/kg) in autumn; (**I**) soil available potassium (mg/kg) in spring; and (**J**) soil available potassium (mg/kg) in autumn. (Note: Treatment abbreviations: RW: Rice–wheat (control); RO: Rice–rape; RV: Rice–hairy vetch; RB: Rice–faba bean; RF: Rice–barley; RS: Rice–fallow. Different letters above bars indicate significant differences between treatments at *p* < 0.05. Error bars represent standard error of three biological replicates (n = 3). Soil samples were collected before rice planting in spring and after rice harvest in autumn).

**Table 1 plants-13-03273-t001:** Seeding rates and spacing for different crops in the rotation system.

Crop	Seeding Rate (kg/ha)	Row Spacing (m)	Plant Spacing (m)
Rice	112.5	0.25	-
Wheat	150	0.25–0.30	-
Barley	187.5	0.25–0.30	-
Rape	-	0.50	0.20
Faba bean	120–150	0.40	0.25
Hairy vetch	60	0.40–0.50	-

Note, (-) indicates not applicable.

**Table 2 plants-13-03273-t002:** Fertilization scheme for different crops in the rotation system (kg/plot).

Crop	Basal Fertilizer			Tillering	Jointing	Heading
	Urea (m^2^)	Ca(H_2_PO_4_)_2_	K_2_SO_4_	Urea	Urea	Urea
Rice	0.83	1.92	0.50	-	1.98	0.44
Wheat	0.84	2.31	0.60	0.984	0.336	-
Barley	0.84	2.31	0.60	0.984	0.336	-
Rape	0.70	2.31	0.60	0.60	0.60	-
Faba bean	0.60	1.92	0.50	-	-	-
Hairy vetch	0.60	1.92	0.50	-	-	-

Note: For rape, tillering fertilizer was applied as seedling fertilizer before winter, and jointing fertilizer was applied as stem-elongation fertilizer after winter. (-) indicates no fertilizer application at that growth stage. Plot size = 48 m^2^. All fertilizer amounts are expressed as kg per plot.

**Table 3 plants-13-03273-t003:** The efficiency of rotation patterns on crop biomass in spring and rice plant biomass.

Rotation	Spring Crop Biomass Efficiency (%)	Rice Plant Biomass Efficiency (%)
Rice–Wheat (S1)	-	-
Rice–Rape (S2)	16.67%	15.00%
Rice–Hairy Vetch (S3)	25.00%	20.00%
Rice–Barley (S4)	8.33%	5.00%
Rice–Faba Bean (S5)	33.33%	30.00%
Rice–Winter Fallow (S6)	−8.33%	−10.00%

Note, the rice–wheat (S1) was set as the control in the efficiency calculation.

## Data Availability

Due to privacy issues, the data presented in this study are available on request from the corresponding author.

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
