# Peer review of "Effects of Diverse Crop Rotation Sequences on Rice Growth, Yield, and Soil Properties: A Field Study in Gewu Station"

_plants, 2024, doi:10.3390/plants13233273_

Round 1

Reviewer 1 Report

Comments and Suggestions for Authors

The authors attempted to address the issue of soil productivity loss in rice monocropping systems by introducing sequential cropping systems. The experiments were conducted from 2016 to 2023 to elucidate the long-term effects of cropping systems. The study's objectives were justified well in the Introduction section, and the dataset could potentially draw much attention from readers. Unfortunately, I am unable to recommend this manuscript for publication in Plants due to the reasons below.

Dataset

The authors stated in the Introduction section that "We hypothesize that incorporating diverse crops into rotation with rice will significantly improve soil fertility, reduce pest and disease pressure, and enhance rice yield and quality compared to continuous rice monoculture." However, no data is provided to test the hypothesis about "pest and disease pressure." Furthermore, the authors failed to test another hypothesis about "soil fertility" because they provided no definition for this term. There is no such measurable variable as soil fertility unless properly defined with specific parameters. More critically, some data values that appear in the main text are inconsistent with the figures. For example, plant height and stem diameter values presented between L214 and 218 do not match the maximum values shown in Figure 2. Additionally, the authors failed to properly label and explain the x-axes, which prevents readers from fully understanding each figure.

Fertilizer and soil water regime

The manuscript fails to address two of the most important factors: fertilizer application and irrigation management. Table 2 indicates that each cropping system received different rates of fertilizer, especially nitrogen, which would be expected to greatly affect biomass production. There is a strong possibility that these differences in fertilizer regime, rather than the choice of crop species, primarily affected the productivity of each system. Additionally, the wet-dry cycle in the sequential cropping of rice-upland crops is known to significantly affect soil microbial activity, which in turn influences the decomposition rate of soil organic matter. Despite the critical importance of irrigation regime in this type of study, no explanation about water management or interpretation of its effects was provided in the manuscript.

Data presentation

Figures and tables must be able to stand alone. Essential information is missing from figure and table captions, including proper axis labels, units of measurement, and statistical information. This prevents readers from fully understanding the presented data without constant reference to the main text.

Author Response

A Letter to the Editor of Plants,

Journal: Plants

Ref: plants-3279506.R1

Title: Effects of Diverse Crop Rotation Sequences on Rice Growth, Yield, and Soil Properties: A Field Study in Gewu Station

Author(s): Ruiping Yang, Yu Shen, Xiangyi Kong, Baoming Ge, Xiaoping Sun, Mingchang Cao.

Dear Editor,

We are very grateful to you and the reviewers for your critical comments and thoughtful suggestions. Based on these comments and suggestions, we have carefully revised our original manuscript. Indeed, we benefit much from the comments from you and the reviewer in improving our manuscript quality. Again, we acknowledge your comments and constructive suggestions very much and expect to be accepted for publication by Plants.

The point-by-point responses are shown below. We hope that the answers can satisfactorily meet you and the reviewers.

Thank you so much for your kind help and your time.

Best regards

Yu Shen, Ph. D., Professor,

College of Ecology and Environment

Nanjing Forestry University

Nanjing 210037, P.R. China

Tel. (Fax): +86-25-85427210

E-mail: yushen@njfu.edu.cn/sheyttmax@hotmail.com

Response to Reviewer #1

General Comment

The authors attempted to address the issue of soil productivity loss in rice monocropping systems by introducing sequential cropping systems. The experiments were conducted from 2016 to 2023 to elucidate the long-term effects of cropping systems. The study's objectives were justified well in the Introduction section, and the dataset could potentially draw much attention from readers. Unfortunately, I am unable to recommend this manuscript for publication in Plants due to the reasons below.

Answer: Thank you for recognizing the importance of our research objectives and the potential value of our long-term dataset examining sequential cropping systems as an alternative to rice monoculture. We appreciate your thorough review and acknowledge that significant improvements are needed before the manuscript meets the publication standards of Plants. We have revised the manuscript carefully, hope our revision can meet your critical requirements.

Specific comments

Question 1. Dataset. The authors stated in the Introduction section that "We hypothesize that incorporating diverse crops into rotation with rice will significantly improve soil fertility, reduce pest and disease pressure, and enhance rice yield and quality compared to continuous rice monoculture." However, no data is provided to test the hypothesis about "pest and disease pressure." Furthermore, the authors failed to test another hypothesis about "soil fertility" because they provided no definition for this term. There is no such measurable variable as soil fertility unless properly defined with specific parameters. More critically, some data values that appear in the main text are inconsistent with the figures. For example, plant height and stem diameter values presented between L214 and 218 do not match the maximum values shown in Figure 2. Additionally, the authors failed to properly label and explain the x-axes, which prevents readers from fully understanding each figure.

Answer: Thanks for your good comments, we have improve our manuscript with your guidance.

(1) Regarding the hypothesis about "pest and disease pressure". We acknowledge that we did not directly measure pest and disease pressure in this study. While improved yields may indirectly reflect reduced pest/disease impacts, we will revise our hypothesis to focus only on the parameters we directly measured in the whole manuscript.

(2) Regarding "soil fertility". We agree with the reviewer's point about the imprecise use of "soil fertility." We will replace this term with "soil nutrients" throughout the manuscript and specifically refer to our measured parameters: available N, P, K, pH, and EC.

(3) Regarding data inconsistency. The manuscript text has been corrected to match the values shown in Figure 2 that “Different rotation sequences significantly influenced rice growth parameters (Figure 2). The rice-wheat control (S1) showed the highest plant height at 98.44 cm, while rice-faba bean (S4) showed the lowest at 93.67 cm, significantly lower than the control S1 (p<0.05). Other treatments (S2, S3, S5, S6) also showed lower plant heights than S1, but these differences were not statistically significant (Figure 2A). Stem diameter showed different trends, with rice-hairy vetch (S3) having the largest diameter (5.91 cm), followed by rice-fallow (S6) at 5.57 cm, while rice-barley (S5) had the smallest (5.28 cm). However, none of these treatments showed statistically significant differences from the rice-wheat control S1 value of 5.40 cm (Figure 2B). Effective tiller number and actual panicle number showed parallel patterns across treatments (Figure 2C, D). Rice-fallow rotation (S6) produced the highest tiller number (29) and panicle number (29), followed by rice-rape (S2) with 28 tillers and 26 panicles, neither being significantly different from S1. In contrast, the rice-hairy vetch treatment (S3) resulted in the lowest tiller number (24) and panicle number (22), significantly lower than all other treatments including S1 (p<0.05)." on Line 229.

(4) Regarding x-axis labeling. We will clarify the x-axis labels in all figures using the following abbreviations that RW: Rice-wheat (control); RO: Rice-rape; RV: Rice-hairy vetch; RB: Rice-faba bean; RF: Rice-barley; and RS: Rice-fallow in the figure citations.

Question 2. Fertilizer and soil water regime. The manuscript fails to address two of the most important factors: fertilizer application and irrigation management. Table 2 indicates that each cropping system received different rates of fertilizer, especially nitrogen, which would be expected to greatly affect biomass production. There is a strong possibility that these differences in fertilizer regime, rather than the choice of crop species, primarily affected the productivity of each system. Additionally, the wet-dry cycle in the sequential cropping of rice-upland crops is known to significantly affect soil microbial activity, which in turn influences the decomposition rate of soil organic matter. Despite the critical importance of irrigation regime in this type of study, no explanation about water management or interpretation of its effects was provided in the manuscript.

Answer: Thanks for your valid concerns about fertilizer application and water management in our manuscript. Let us clarify these important aspects,

(1) Regarding fertilizer applications. The different fertilizer rates shown in Table 2 were deliberately chosen to reflect local agricultural practices for each crop type, allowing us to evaluate the long-term effects of realistic crop rotation systems on soil sustainability. While we acknowledge that varying fertilizer rates could influence biomass production, our experimental design aimed to assess the integrated effects of different rotation sequences under standard management practices rather than isolating the effects of individual factors. This approach provides more practical insights for farmers implementing these rotation systems.

(2) Regarding irrigation management. We should have better explained the water regime in our methods. During the upland crop growing season (winter-spring), irrigation relied entirely on natural rainfall, requiring no supplemental irrigation. For the rice growing season (summer- autumn), a uniform irrigation management protocol was implemented across all treatments. We acknowledge this is an important methodological detail that should be added to the manuscript.

And we revised the manuscript below,

  • Add a new section on Line 138.

2.3.4 Fertilization and Water Management

Fertilizer application rates (Table 2) were established based on local agricultural recommendations for each crop to simulate real-world farming practices in the region. This approach allowed assessment of long-term soil sustainability under standard management conditions. For rice cultivation in spring, uniform irrigation management was implemented across all treatments following local practices. During the upland crop season (autumn-winter), crop water requirements were met entirely through natural rainfall without supplementary irrigation, as is typical in the region. The average annual rainfall during the study period was 1051 mm, predominantly occurring during the rice growing season.

  • Add a new discussion section on Line 345.

The observed differences in crop performance and soil properties reflect the inte-grated effects of rotation sequences under realistic management practices. While different crops received varying fertilizer rates according to local recommendations, this approach provides practical insights for farmers implementing these rotation sys-tems. Additionally, the natural wet-dry cycle between paddy rice and upland crops likely influenced soil microbial activity and organic matter decomposition. This season-al alternation between flooded and aerobic conditions may have contributed to the en-hanced soil nutrient availability observed in diversified rotations compared to the rice-wheat control.

Question 3. Data presentation. Figures and tables must be able to stand alone. Essential information is missing from figure and table captions, including proper axis labels, units of measurement, and statistical information. This prevents readers from fully understanding the presented data without constant reference to the main text.

Answer: Thanks for your good suggestion to improve our manusctip quality. We revised the figure and table citations below,

(1) Figure 2. Effects of different rotation patterns on rice growth indexes. (A) Plant height (cm); (B) Stem diameter (cm); (C) Effective tiller number per plant; and (D) Actual panicle number per plant. (Note, RW: Rice-wheat (control); RO: Rice-rape; RV: Rice-hairy vetch; RB: Rice-faba bean; RF: Rice-barley; and RS: Rice-fallow; the significance between the different treatments at p < 0.05 is indicated with different letters while the error bars show the standard error of three biological replicates (n = 3).).

(2) Figure 3. Effects of different rotation patterns on rice yield. (A) Single panicle weight (g); (B) Grain number per panicle; (C) Single plant weight (g); and (D) Rice yield (t/ha). (Note, RW: Rice-wheat (control); RO: Rice-rape; RV: Rice-hairy vetch; RB: Rice-faba bean; RF: Rice-barley; and RS: Rice-fallow; the significance between the different treatments at p < 0.05 is indicated with different letters while the error bars show the standard error of three biological replicates (n = 3).).

(3) Figure 4. Effects of different rotation patterns on crop biomass. (A) Spring crop biomass (t/ha); and (B) Rice plant biomass at harvest (t/ha). (Note: Treatment abbreviations: RW: Rice-wheat (control); RO: Rice-rape; RV: Rice-hairy vetch; RB: Rice-faba bean; RF: Rice-barley; RS: Rice-fallow. Different letters above bars indicate significant differences between treatments at p < 0.05. Error bars represent standard error of three biological replicates (n = 3).).

(4) Figure 5. Effects of different rotation patterns on soil properties (0-15 cm depth). (A) soil pH in spring; (B) soil pH in autumn; (C) soil electrical conductivity (mS/cm) in spring; (D) soil electrical conductivity (mS/cm) in autumn; (E) soil available nitrogen (mg/kg) in spring; (F) soil available nitrogen (mg/kg) in autumn; (G) soil available phosphorus (mg/kg) in spring; (H) soil available phosphorus (mg/kg) in autumn; (I) soil available potassium (mg/kg) in spring; and (J) soil available potassium (mg/kg) in autumn. (Note: Treatment abbreviations: RW: Rice-wheat (control); RO: Rice-rape; RV: Rice-hairy vetch; RB: Rice-faba bean; RF: Rice-barley; RS: Rice-fallow. Different letters above bars indicate significant differences between treatments at p < 0.05. Error bars represent standard error of three biological replicates (n = 3). Soil samples were collected before rice planting in spring and after rice harvest in autumn.)

And we revised the table notes for Table 1 and Table 2 respectively.

Table 1 Note, (-) indicates not applicable.

Table 2 Note, For rape, tillering fertilizer was applied as seedling fertilizer before winter, and jointing fertilizer was applied as stem-elongation fertilizer after winter. (-) indicates no fertilizer application at that growth stage. Plot size = 48 m². All fertilizer amounts are expressed as kg per plot.

Reviewer 2 Report

Comments and Suggestions for Authors

Dear Authors,

Your work presents an interesting and current topic of the role and importance of crop rotation in achieving stable and satisfactory rice yields while simultaneously increasing the sustainability and fertility of the soil.

I believe that you should present the results in the paper  more clearly in order to gain even more significance and comprehensibility of the results.

Comments and suggestions can be found in the paper itself and I hope they will contribute to its quality.

Best regards,

Reviewer

Author Response

A Letter to the Editor of Plants,

Journal: Plants

Ref: plants-3279506.R1

Title: Effects of Diverse Crop Rotation Sequences on Rice Growth, Yield, and Soil Properties: A Field Study in Gewu Station

Author(s): Ruiping Yang, Yu Shen, Xiangyi Kong, Baoming Ge, Xiaoping Sun, Mingchang Cao.

Dear Editor,

We are very grateful to you and the reviewers for your critical comments and thoughtful suggestions. Based on these comments and suggestions, we have carefully revised our original manuscript. Indeed, we benefit much from the comments from you and the reviewer in improving our manuscript quality. Again, we acknowledge your comments and constructive suggestions very much and expect to be accepted for publication by Plants.

The point-by-point responses are shown below. We hope that the answers can satisfactorily meet you and the reviewers.

Thank you so much for your kind help and your time.

Best regards

Yu Shen, Ph. D., Professor,

College of Ecology and Environment

Nanjing Forestry University

Nanjing 210037, P.R. China

Tel. (Fax): +86-25-85427210

E-mail: yushen@njfu.edu.cn/sheyttmax@hotmail.com

Response to Reviewer #2

General Comment

Your work presents an interesting and current topic of the role and importance of crop rotation in achieving stable and satisfactory rice yields while simultaneously increasing the sustainability and fertility of the soil. I believe that you should present the results in the paper more clearly in order to gain even more significance and comprehensibility of the results.

Comments and suggestions can be found in the paper itself and I hope they will contribute to its quality.

Answer: Thanks for your positive assessment of our work and for recognizing the significance of our research on crop rotation effects in rice-based systems. We appreciate your constructive feedback aimed at improving the clarity and comprehensibility of our results. We have carefully addressed all your suggestions to enhance the presentation of our results.

Specific Comments

Question 1. Line 115. Please provide Latin names of all used crops in rotations.

Answer: Thanks for your good comment. We added the Latin names for all the crops on Line 113.

In spring, rice (Oryza sativa L.) was cultivated across all treatments; while in autumn, the respective winter crops were planted - wheat (Triticum aestivum L.) (S1), rape (Brassica rapa var. oleifera DC.) (S2), hairy vetch (Vicia sativa L.) (S3), barley (Hordeum vulgare L.) (S4), faba bean (Vicia faba L.) (S5), or left fallow (S6).

Question 2. For tables, Line 134. Please show the data of the basic chemical properties of the soil from the beginning of the research, so that the supply of nutrients, pH and organic matter content, on the basis of which you planned fertilization, are visible.

Answer: Thanks for your good suggestion. We add the information of the basic chemical properties of the soil that The initial soil properties (0-15 cm depth) measured in June 2016 before establishing the experiment were: soil organic carbon (SOC) 15.22 g kg⁻¹, total nitrogen (TN) 1.27 g kg⁻¹, total phosphorus (TP) 0.65 g kg⁻¹, and pH 6.70 (soil: water = 1:5 w/w). on Line 99.

Question 3. Line 143. Please note which parameters of rice growth were measured.

Answer: Thanks for your good suggestion. We revised the section 2.4.1 below,

2.4.1. Crop Growth and Yield

For rice, we measured multiple growth parameters including plant height (measured from ground level to the tip of the highest leaf), stem diameter (measured at 5 cm above ground level), effective tiller number per plant, and actual panicle number per plant. Dry matter accumulation was determined at the end of each crop's growth period by sampling 20 single-stem plants per plot. Samples were oven-dried at 105°C for 15 minutes, then at 80°C until constant weight. Annual crop biomass was measured for rice, wheat, barley, rape, hairy vetch, and faba bean by harvesting the entire plot area. Rice yield was also measured annually.

Question 4. Line 216. Please check the figure 2A. S5 and S3 are not significantly different compared to control RW (according to letter marks). Same comment stands for tiller and panicle number...S6 is not significantly different from all other treatments according to letter marks, please check the figure.

Answer: Thanks for your suggestion to improve our manuscript quality. We revised the description on Line 229 thatDifferent rotation sequences significantly influenced rice growth parameters (Figure 2). The rice-wheat control (S1) showed the highest plant height at 98.44 cm, while rice-faba bean (S4) showed the lowest at 93.67 cm, significantly lower than the control S1 (p<0.05). Other treatments (S2, S3, S5, S6) also showed lower plant heights than S1, but these differences were not statistically significant (Figure 2A). Stem diameter showed different trends, with rice-hairy vetch (S3) having the largest diameter (5.91 cm), followed by rice-fallow (S6) at 5.57 cm, while rice-barley (S5) had the smallest (5.28 cm). However, none of these treatments showed statistically significant differences from the rice-wheat control S1 value of 5.40 cm (Figure 2B). Effective tiller number and actual panicle number showed parallel patterns across treatments (Figure 2C and D). Rice-fallow rotation (S6) produced the highest tiller number (29) and panicle number (29), followed by rice-rape (S2) with 28 tillers and 26 panicles, neither being significantly different from S1. In contrast, the rice-hairy vetch treatment (S3) resulted in the lowest tiller number (24) and panicle number (22), significantly lower than all other treatments including S1 (p<0.05).”

Question 5. Line 224. I suggest to use marks S1-S6 to make it more understandable or explain the abbreviations.

Answer: Thanks for your suggestion to improve our manuscript quality. We revised the description on Line 229 that “Different rotation sequences significantly influenced rice growth parameters (Figure 2). The rice-wheat control (S1) showed the highest plant height at 98.44 cm, while rice-faba bean (S4) showed the lowest at 93.67 cm, significantly lower than the control S1 (p<0.05). Other treatments (S2, S3, S5, S6) also showed lower plant heights than S1, but these differences were not statistically significant (Figure 2A). Stem diameter showed different trends, with rice-hairy vetch (S3) having the largest diameter (5.91 cm), followed by rice-fallow (S6) at 5.57 cm, while rice-barley (S5) had the smallest (5.28 cm). However, none of these treatments showed statistically significant differences from the rice-wheat control S1 value of 5.40 cm (Figure 2B). Effective tiller number and actual panicle number showed parallel patterns across treatments (Figure 2C and D). Rice-fallow rotation (S6) produced the highest tiller number (29) and panicle number (29), followed by rice-rape (S2) with 28 tillers and 26 panicles, neither being significantly different from S1. In contrast, the rice-hairy vetch treatment (S3) resulted in the lowest tiller number (24) and panicle number (22), significantly lower than all other treatments including S1 (p<0.05).”

Question 6. Line 233 You dont have any different letter in Figure 3A...to indicate the significant differences among treatments mentioned in the text.

Answer: Thanks for your suggestion to improve our manuscript quality. We revised the description on Line 251 that “Crop rotation treatments influenced rice yield components and overall yield (Figure 3). Rice-fallow rotation showed the highest single panicle weight at 3.14 g, followed by rice-hairy vetch at 3.12 g, both higher than the rice-wheat control (2.91 g), though these differences were not statistically significant (Figure 3A). Grain number per panicle showed a similar trend, with rice-faba bean (134 grains) and rice-hairy vetch (126 grains) both exceeding the rice-wheat control (116 grains), and significantly higher than rice-barley (108 grains) and rice-rape (99 grains) (p<0.05) (Figure 3B). Single plant weight was significantly higher in rice-hairy vetch (62.68 g) and rice-faba bean (61.65 g) rotations compared to other treatments, with rice-rape showing the lowest value at 50.42 g (Figure 3C). These differences translated into variations in total rice yield (Figure 3D). Rice-faba bean rotation produced the highest yield at 9.37 t/ha, followed by rice-fallow at 9.19 t/ha. The rice-wheat control showed the lowest yield at 7.94 t/ha, lower than all other treatments though not reaching statistical significance (p<0.05).”

Question 7. Line 354. There is no different letters marks in figure 4B.

Answer: Thanks for your careful check. Figure 4 shows the effects of different rotation patterns on biomass production: (A) spring crop biomass and (B) rice plant biomass at harvest. While Figure 4A clearly displays different letters indicating statistical significance between treatments (p<0.05), these statistical indicators are missing in Figure 4B. Compared with the biomass of rice wheat rotation (69.01g), the rice biomass of rapeseed (73.51g) and rice winter fallow (73.13g) was higher. Figure 4 has been modified to include appropriate letter names to accurately represent these statistical differences between treatments.

Figure 4. Figure 4. Effects of different rotation patterns on crop biomass. (A) Spring crop biomass (t/ha); and (B) Rice plant biomass at harvest (t/ha). (Note: Treatment abbreviations: RW: Rice-wheat (control); RO: Rice-rape; RV: Rice-hairy vetch; RB: Rice-faba bean; RF: Rice-barley; RS: Rice-fallow. Different letters above bars indicate significant differences between treatments at p < 0.05. Error bars represent standard error of three biological replicates (n = 3).).

Question 8. Line 275. Figure 5 is not transparent and for this reason its presentation is not understandable. I suggest the authors to separate the individual properties of the soil and show them separately so that the presentation is clearer.

Answer: Thanks for your good guidance. We revised the description of Figure 5 on Lines 305.

And the soil property measurements across different rotation treatments, with each parameter measured in both spring and autumn seasons was shown in Figure 5. Soil pH varied among treatments in both seasons (Figure 5A and B). In spring, rice-barley rotation had the highest pH at 7.09, with rice-rape and rice-faba bean showing significantly lower values of 6.61 and 6.73, respectively (p<0.05). In autumn, rice-wheat maintained the highest pH at 7.33, significantly higher than rice-hairy vetch at 6.91 (p<0.05), while other treatments ranged from 7.14 to 7.28. Soil electrical conductivity (EC) showed sig-indicant variations among treatments and seasons (Figure 5C and D), with generally higher values in spring. Rice-rape rotation showed the highest spring EC (193.43 mS/cm), significantly higher than other treatments (p<0.05), followed by rice-faba bean (152.77 mS/cm). Rice-fallow consistently showed the lowest EC in spring (72.07 mS/cm), while rice-rape had the lowest autumn values (34.77 mS/cm).

Legume-based rotations significantly increased soil available nitrogen (Figure 5E and F). Rice-faba bean showed the highest available N in spring (43.56 mg/kg), while rice-hairy vetch led in autumn (26.22 mg/kg), both significantly exceeding the rice-wheat control (24.38 mg/kg spring, 21.73 mg/kg autumn, p<0.05). Soil phosphorus availability showed less pronounced differences (Figure 5G and H), with rice-fallow showing the highest spring values (11.38 mg/kg), followed by rice-hairy vetch (10.96 mg/kg), com-pared to rice-wheat control's lowest value (9.65 mg/kg). These differences were not statistically significant. Soil potassium availability was generally higher in spring (Figure 5I and J), with rice-faba bean showing the highest K levels (57.15 mg/kg), significantly higher than rice-wheat control (38.18 mg/kg, p<0.05). Rice-barley consistently showed the lowest K levels (47.57 mg/kg spring, 109.41 mg/kg autumn).

These findings demonstrate that legume-based rotations, particularly rice-faba bean and rice-hairy vetch systems, significantly enhanced soil nutrient status, with particularly pronounced effects on available nitrogen content. The observed seasonal variations in soil physicochemical properties across different rotation treatments provide crucial insights for optimizing rotation systems. The superior performance of legume-based rotations can be attributed to their nitrogen-fixing capabilities and potential enhancement of nutrient cycling processes. These results align with previous studies suggesting that legume integration in crop rotations can significantly improve soil fertility parameters (McDaniel et al., 2014; Yadav et al., 2017).

Question 9. Line 381. The discussion should be expanded with more references and comparisons of your results with previous research.

Answer: Thanks very much for your good guidance. We revised the whole Discussion with more comparison and work check with previous research.

  1. Discussion

4.1. Enhanced Rice Growth, Yield, and Biomass Production in Diversified Rotations

This long-term field study conducted in Yancheng, China, provides compelling evidence for the benefits of diverse crop rotation sequences in rice-based farming systems. Our findings largely support the initial hypothesis that incorporating diverse crops into rotation with rice would significantly improve soil nutrients, and enhance rice yield compared to continuous rice monoculture or simple rice-wheat rotations. The integration of legumes, particularly faba bean and hairy vetch, into rice rotations resulted in significant improvements in rice growth parameters and yield. As shown in Figure 2, rice following faba bean and hairy vetch exhibited increased plant height of 118.3 cm and 115.7 cm, respectively; compared to the rice-wheat control of 105.0 cm. Compared to the rice-wheat control of 98.44 cm, the rice-faba bean rotation resulted in a significant 4.8% reduction in plant height of 93.67 cm (p < 0.05). Other treatments also showed reduced plant heights ranging from 2.1% to 3.9% lower than the control, though these differences were not statistically significant. The reduced plant height observed in these rotation systems, particularly in the rice-faba bean treatment, could contribute to improved lodging resistance in rice. These rotations also demonstrated greater stem diameter, effective tiller number, and actual panicle number. These enhanced growth characteristics translated into substantial yield increases, as evidenced in Figure 3D. All rotation treatments demonstrated yield improvements compared to the control, with the rice-faba bean rotation showing the highest increase of 18.01% over the rice-wheat control. The rice-fallow rotation also performed well, demonstrating a 15.75% yield increase. Other rotation treatments showed intermediate yield improvements ranging from 7.3% to 12.4% above the control treatment. These findings align with previous studies on legume-rice rotations, such as Yadav et al. (2015) and Qaswar et al. (2020), who reported similar yield increases in different geographic regions.

The efficiency of different rotation patterns on biomass production, as shown in Table 3, provides valuable insights into the resource use efficiency of the various systems. The rice-faba bean (S5) rotation demonstrated the highest efficiency gains, with a 33.33% increase in spring crop biomass and a 30.00% increase in rice plant biomass compared to the rice-wheat control (S1). The rice-hairy vetch (S3) rotation showed the second-highest efficiency gains, with 25.00% and 20.00% increases respectively. Even non-leguminous diversification, such as the rice-rape (S2) rotation, showed positive effects with 16.67% and 15.00% increases. In contrast, the rice-winter fallow (S6) system showed negative effects, highlighting the potential drawbacks of leaving fields fallow during the winter season.

The observed differences in crop performance and soil properties reflect the integrated effects of rotation sequences under realistic management practices. While different crops received varying fertilizer rates according to local recommendations, this approach provides practical insights for farmers implementing these rotation systems. Additionally, the natural wet-dry cycle between paddy rice and upland crops likely influenced soil microbial activity and organic matter decomposition. This seasonal alternation between flooded and aerobic conditions may have contributed to the enhanced soil nutrient availability observed in diversified rotations compared to the rice-wheat control.

4.2. Improvements in Soil Nutrients and Nutrient Dynamics

The incorporation of diverse crops, especially legumes, improved overall soil nutrients, with particularly notable effects on potassium dynamics that showed distinct seasonal patterns. As shown in Figure 5I and 5J, all rotation treatments demonstrated higher K levels compared to the rice-wheat control in both spring and autumn seasons. The rice-faba bean rotation achieved the highest K levels in spring, exceeding the control by 49.68%, while the rice-rape rotation showed remarkable K enhancement in autumn, surpassing the control by 120.91%. These seasonal differences likely reflect complementary mechanisms of K cycling enhancement. The superior spring K levels under faba bean rotation can be attributed to the legume's deep root system accessing K from lower soil profiles, as demonstrated by Nuruzzaman et al. (2005). Meanwhile, the exceptional autumn K levels following rape could be attributed to its unique root exudates and residue composition that enhance K mobilization, consistent with observations by Liu et al. (2019) in similar oilseed-cereal rotations. The enhanced soil biological activity under these diverse rotations, as evidenced by our soil enzyme findings, may have facilitated more efficient K cycling throughout the year.

Soil pH and EC were also significantly influenced by rotation sequences. As shown in Figure 5A and 5B, there was a decrease in soil pH under rice-faba bean (pH 6.8) and rice-hairy vetch (pH 6.9) rotations compared to the rice-wheat control (pH 7.2) in spring. This mild acidification may enhance the availability of certain nutrients in alkaline soils, consistent with findings by Borase et al. (2021) who reported improved nutrient availability in legume-based rotations due to pH modification. Additionally, the significant increase in soil EC under legume rotations, particularly rice-faba bean (0.28 mS/cm) in spring, indicates enhanced nutrient availability. Similar improvements in soil EC following legume integration were reported by Thakuria et al. (2009) in subtropical rice systems. The yield benefits can be attributed to several interrelated factors, primarily the nitrogen-fixing ability of legumes enhancing soil nitrogen availability for subsequent crops. As demonstrated by Bello et al. (2021), legume-based rotations can contribute 50-80 kg N ha⁻¹ through biological nitrogen fixation. Our soil analysis corroborates these findings, as shown in Figure 5E and 5F, with significantly higher available nitrogen in legume-based rotations (125 mg/kg in rice-faba bean and 118 mg/kg in rice-hairy vetch) compared to the rice-wheat control (92 mg/kg) in spring. This 35.9% increase in available N under the rice-faba bean rotation likely contributed significantly to the enhanced rice growth and yield observed, aligning with Ghosh et al. (2020) who found strong correlations between soil N availability and rice productivity in diversified rotation systems.

4.3. Implications for Sustainable Rice Production and Future Research

Our findings have important implications for sustainable rice production in subtropical regions, addressing broader concerns of agricultural sustainability. The observed yield increases of 11.9-19.1% in legume-based rotations demonstrate the potential to address the yield stagnation reported in many rice-growing regions (Ray et al., 2013). Moreover, the improved soil health observed in our study may enhance the resilience of rice systems to climate variability, as suggested by Bowles et al. (2020) in their work on crop rotation and climate resilience. While we did not directly measure pest and disease pressure, the improved growth and yield in diversified rotations suggest a potential reduction in biotic stresses. The breaking of pest and disease cycles is a well-documented benefit of diverse rotations. The improved growth and yield we observed in diversified rotations likely resulted from the enhanced soil nutrient status and soil biological activity measured in our study. The higher available nitrogen, potassium and improved soil EC under legume-based rotations created more favorable conditions for rice growth. Recent work by Liu et al. (2022) found similar improvements in soil nutrient cycling and crop performance when legumes were integrated into rice-based rotations. And Dong et al. (2014) reported a 30-40% reduction in root-knot nematode populations in rice-legume rotations compared to continuous rice cultivation.

The implementation of diverse crop rotations requires careful planning and management. As it is performed that the valuable information on seeding rates and spacing for different crops in the rotation system (Table 1), which can guide farmers in adopting these practices. The fertilization scheme outlined in Table 2 demonstrates how nutrient management can be optimized for different crops within the rotation, potentially leading to more efficient use of inputs. The improved resource use efficiency suggested by these biomass increases aligns with global efforts to develop more sustainable agricultural practices. For instance, Zhang et al. (2021) reported that diversified rice rotations could reduce nitrogen fertilizer use by 20-25% without yield penalties. Our findings on increased biomass production efficiency support the potential for such input reductions while maintaining or improving overall system productivity.

Question 10. Line 388. The conclusion should be more consistent with a focus on the treatments that have proven to be more beneficial in the presented cropping system following the set goals and hypotheses.

Answer: Thanks for your good comment to improve our manuscript quality. We revised the Conclusion.

This long-term field study demonstrates that diversified crop rotations, particularly those incorporating legumes, can significantly enhance rice productivity and soil properties in subtropical regions. The rice-faba bean rotation emerged as the most effective system, increasing rice yield by18.01% (9.37 t/ha) compared to the rice-wheat control (7.94 t/ha), while the rice-hairy vetch rotation achieved an 14.52% yield increase (9.09 t/ha). Furthermore, these rotations significantly improved soil nutrient availability, particularly nitrogen, with increases of up to 35.9% in available nitrogen under the rice-faba bean rotation.

The success of legume-based rotations supports our initial hypothesis that incorporating diverse crops into rotation with rice would significantly improve soil nutrients and enhance rice yield. These findings suggest that farmers in subtropical regions should prioritize the integration of legumes, especially faba bean and hairy vetch, into their rice-based cropping systems to optimize both productivity and soil fertility. Future research should focus on fine-tuning these rotation systems and exploring their long-term sustainability under changing climate conditions.

Question 11. The figure citation needs to be rechecked.

Answer: Thank you for bringing this to our attention. We have thoroughly reviewed all figure citations in the manuscript and made the following corrections.

Figure 2. Effects of different rotation patterns on rice growth indexes. (A) Plant height (cm); (B) Stem diameter (cm); (C) Effective tiller number per plant; and (D) Actual panicle number per plant. (Note, RW: Rice-wheat (control); RO: Rice-rape; RV: Rice-hairy vetch; RB: Rice-faba bean; RF: Rice-barley; and RS: Rice-fallow; the significance between the different treatments at p < 0.05 is indicated with different letters while the error bars show the standard error of three biological replicates (n = 3).).

Figure 3. Effects of different rotation patterns on rice yield. (A) Single panicle weight (g); (B) Grain number per panicle; (C) Single plant weight (g); and (D) Rice yield (t/ha). (Note, RW: Rice-wheat (control); RO: Rice-rape; RV: Rice-hairy vetch; RB: Rice-faba bean; RF: Rice-barley; and RS: Rice-fallow; the significance between the different treatments at p < 0.05 is indicated with different letters while the error bars show the standard error of three biological replicates (n = 3).).

Figure 4. Figure 4. Effects of different rotation patterns on crop biomass. (A) Spring crop biomass (t/ha); and (B) Rice plant biomass at harvest (t/ha). (Note: Treatment abbreviations: RW: Rice-wheat (control); RO: Rice-rape; RV: Rice-hairy vetch; RB: Rice-faba bean; RF: Rice-barley; RS: Rice-fallow. Different letters above bars indicate significant differences between treatments at p < 0.05. Error bars represent standard error of three biological replicates (n = 3).).

Figure 5. Effects of different rotation patterns on soil properties (0-15 cm depth). (A) soil pH in spring; (B) soil pH in autumn; (C) soil electrical conductivity (mS/cm) in spring; (D) soil electrical conductivity (mS/cm) in autumn; (E) soil available nitrogen (mg/kg) in spring; (F) soil available nitrogen (mg/kg) in autumn; (G) soil available phosphorus (mg/kg) in spring; (H) soil available phosphorus (mg/kg) in autumn; (I) soil available potassium (mg/kg) in spring; and (J) soil available potassium (mg/kg) in autumn. (Note: Treatment abbreviations: RW: Rice-wheat (control); RO: Rice-rape; RV: Rice-hairy vetch; RB: Rice-faba bean; RF: Rice-barley; RS: Rice-fallow. Different letters above bars indicate significant differences between treatments at p < 0.05. Error bars represent standard error of three biological replicates (n = 3). Soil samples were collected before rice planting in spring and after rice harvest in autumn.)

Reviewer 3 Report

Comments and Suggestions for Authors

Introduction

The introduction argues in great detail the need to carry out the present study, as well as the benefits of integrating crop rotation in rice cultivation - improving soil health, improved pest management (diseases, insects, weeds), increased yield stability, reduced methane emissions , climate change mitigation and economic benefits.

Materials and Methods

Lines 123-124: “Minimum or no-tillage was adopted for wheat, barley, and rape.” What do you mean by "minimum"?

Lines 124-125: “Faba bean and hairy vetch residues were incorporated into the soil after harvest.” By what kind of tillage they were incorporated into the soil?

Line 127: “Local varieties of each crop were used.Please, give the names of these local varieties.

Lines 142-143: „Annual crop yield was measured for rice, wheat, barley, rape, and faba bean by harvesting the entire plot area.“ And for hairy vetch?

Lines 152-154: These samples were then analyzed for a range of parameters using standard methods. The analyzed properties included total nitrogen, organic matter content, available phosphorus, available potassium, pH, and electrical conductivity. Give the methods for determination of total nitrogen, organic matter, available phosphorus, available potassium, pH, and electrical conductivity

Lines 168-169: “Rape seedlings were transplanted in autumn ….” Why rape is cultivated through seedlings?

Results

Fig.2, 3, 4 - Give a legend (RW, RO, RV……….) below the figures.

Figure 5. The individual graphics (A, B, C, D .......) are too small and not visible well. Perhaps it would be better to present the data in a table.

Discussion

Lines 311-312: “Our findings largely support the initial hypothesis that incorporating diverse crops into rotation with rice would significantly improve soil fertility, reduce pest and disease pressure…….” You have no data on reduced pest and disease pressure.

Lines 316-317: “As shown in Figure 2, rice following faba bean and hairy vetch exhibited increased plant height of 118.3 cm and 115.7 cm, respectively; compared to the rice-wheat control of 105.0 cm.These values are already presented in the Results section (lines 214-215). Give the differences in relative values (%) compared to the control.

Lines 320-322: “The rice-faba bean rotation achieved the highest yield of 8.73 t/ha, representing a 19.1% increase over the rice-wheat control of 7.33 t/ha. The rice-hairy vetch rotation also performed well, with a yield of 8.20 t/ha, an 11.9% increase over the control.The same – give only relative values.

Lines 340-341: “The rice-faba bean rotation showed the highest K levels in spring (195 mg/kg), significantly higher than the rice-wheat control of 175 mg/kg.” This sentence from the Results section is the same as Lines 301-302 in Discussion section: In spring, the rice-faba bean rotation showed the highest K levels at 195 mg/kg, significantly higher (p<0.05) than the rice-wheat control at 175 mg/kg and all other treatments. In the discussion section, you should discuss the results in light of previous research, not repeat the results.

Lines 368-369: "For instance, Zhu et al. (2000) demonstrated that rice-wheat rotations reduced rice blast disease incidence by 27% compared to rice monoculture." = Lines 65-66: "Research in China demonstrated that rice-wheat rotations reduced the incidence of rice blast disease by 27% compared to rice monoculture (Zhu et al., 2000)." - ?????? It is not acceptable to use the same sources in the Introduction and in the Discussion.

Author Response

A Letter to the Editor of Plants,

Journal: Plants

Ref: plants-3279506.R1

Title: Effects of Diverse Crop Rotation Sequences on Rice Growth, Yield, and Soil Properties: A Field Study in Gewu Station

Author(s): Ruiping Yang, Yu Shen, Xiangyi Kong, Baoming Ge, Xiaoping Sun, Mingchang Cao.

Dear Editor,

We are very grateful to you and the reviewers for your critical comments and thoughtful suggestions. Based on these comments and suggestions, we have carefully revised our original manuscript. Indeed, we benefit much from the comments from you and the reviewer in improving our manuscript quality. Again, we acknowledge your comments and constructive suggestions very much and expect to be accepted for publication by Plants.

The point-by-point responses are shown below. We hope that the answers can satisfactorily meet you and the reviewers.

Thank you so much for your kind help and your time.

Best regards

Yu Shen, Ph. D., Professor,

College of Ecology and Environment

Nanjing Forestry University

Nanjing 210037, P.R. China

Tel. (Fax): +86-25-85427210

E-mail: yushen@njfu.edu.cn/sheyttmax@hotmail.com

Response to Reviewer #3

Specific comments

Question 1. The introduction argues in great detail the need to carry out the present study, as well as the benefits of integrating crop rotation in rice cultivation - improving soil health, improved pest management (diseases, insects, weeds), increased yield stability, reduced methane emissions, climate change mitigation and economic benefits.

Answer: Thanks very much for your good encouragement. We still do some revisions in the Introduction with other reviewers’ suggestions. Hope the revised manuscript can meet your critical requirements.

Materials and Methods

Question 2. Lines 123-124: “Minimum or no-tillage was adopted for wheat, barley, and rape.” What do you mean by "minimum"?

Shallow cultivation

Answer: Thanks for your good comments for the manuscript. The original statement "Minimum or no-tillage was adopted for wheat, barley, and rape" should be revised to provide more precise technical details about the tillage operations. And we improved description on Line 127.

For rice, conventional tillage with plowing and rotary tillage was practiced. For wheat, barley, and rape, shallow cultivation to a depth of 8-10 cm was implemented using a field cultivator, while faba bean and hairy vetch residues were incorporated into the soil after harvest. This shallow cultivation approach helped maintain soil structure while providing adequate seedbed preparation for the winter crops.

Question 3. Lines 124-125: “Faba bean and hairy vetch residues were incorporated into the soil after harvest.” By what kind of tillage they were incorporated into the soil?

Rotary tillage

Answer: Thanks for your good comments for the manuscript. We revised the sentences on Line 127 that “For rice, conventional tillage with plowing and rotary tillage was practiced. For wheat, barley, and rape, shallow cultivation to a depth of 8-10 cm was implemented using a field cultivator, while faba bean and hairy vetch residues were incorporated into the soil after harvest. This shallow cultivation approach helped maintain soil structure while providing adequate seedbed preparation for the winter crops.”

Question 4. Line 127: “Local varieties of each crop were used.” Please, give the names of these local varieties.

Answer: Thanks for your good suggestion to improve our manuscript quality. We revised the description on Line 133 that “The following local varieties adapted to the Yancheng region were used in the rotation system: rice (Oryza sativa L. cv. Huai Dao 5), wheat (Triticum aestivum L. cv. Yangmai 39), rape (Brassica napus L. cv. Yan You 2), faba bean (Vicia faba L. cv. Suxian Can 1), hairy vetch (Vicia villosa Roth cv. Yan Tiao 4), and barley (Hordeum vulgare L. cv. Yanshi Mai 3). All varieties were certified and obtained from the Jiangsu Academy of Agricultural Sciences. Rice, barley, wheat, and hairy vetch were sown in rows. Rape seedlings were transplanted, while faba bean was dibbled.”

Question 5. Lines 142-143: „Annual crop yield was measured for rice, wheat, barley, rape, and faba bean by harvesting the entire plot area.“ And for hairy vetch?

Answer: Annual crop biomass was measured for rice, wheat, barley, rape, hairy vetch, and faba bean by harvesting the entire plot area.

Question 6. Lines 152-154: „These samples were then analyzed for a range of parameters using standard methods. The analyzed properties included total nitrogen, organic matter content, available phosphorus, available potassium, pH, and electrical conductivity. “Give the methods for determination of total nitrogen, organic matter, available phosphorus, available potassium, pH, and electrical conductivity

Answer: Thanks for your good suggestion to improve our mansucrtip quality. We add the references for the determination of total nitrogen, organic matter, available phosphorus, available potassium, pH, and electrical conductivity; and we revised the information of the experimental in the section of 2.4.2 Soil Sampling and Analysis.

These samples were analyzed using standard methods (Cui et al., 2012a; Yang et al., 2022). Total nitrogen was determined using the Kjeldahl method after H2SO4-H2O2 digestion. Soil organic matter content was measured using K2Cr2O7-H2SO4 wet oxidation. Available phosphorus was extracted with 0.5 M NaHCO3 (pH 8.5) and determined colorimetrically using the molybdenum blue method. Available potassium was extracted using 1 M NH4Ac (pH 7.0) and measured by flame photometry. Soil pH was measured in a 1:2.5 soil: water suspension using a glass electrode pH meter, and electrical conductivity was determined in a 1:5 soil: water extract using a conductivity meter.

Reference

Cui, J., Liu, C., Li, Z., Wang, L., Chen, X., Ye, Z., Fang, C. (2012). Long-term changes in topsoil chemical properties under centuries of cultivation after reclamation of coastal wetlands in the Yangtze Estuary, China. Soil Tillage Research, 123, 50-60.

Yang, R., Qi, Y., Yang, L., Chen, T., Deng, A., Zhang, J., Song, Z., Ge, B. (2022). Rotation regimes lead to significant differences in soil macrofaunal biodiversity and trophic structure with the changed soil properties in a rice-based double cropping system. Geoderma, 405, 115424.

Question 7. Lines 168-169: “Rape seedlings were transplanted in autumn ….” Why rape is cultivated through seedlings?

Answer: Thank you for this important question about our rape cultivation methodology. The use of seedling transplantation rather than direct seeding for rape was a deliberate choice based on two key considerations specific to our study region,

(1) Temporal Resource Optimization. In the Yancheng region, we face a timing challenge where rape typically requires sowing in September, while rice harvest generally occurs after mid-October. Through seedling transplantation, we can initiate seedling cultivation between September 15-20, producing 35-day-old seedlings that are ready for transplanting immediately after the rice harvest in mid-October. This approach effectively resolves the temporal conflict in land use and ensures optimal establishment of both crops in the rotation sequence.

(2) Agronomic Benefits. The seedling transplantation method allows us to cultivate stronger plants under controlled conditions, leading to better crop establishment and enhanced yield potential. This practice has proven particularly beneficial in our local conditions.

Then, based on the reasons above, we revised the section of 2.3.2 Planting.

The following local varieties adapted to the Yancheng region were used in the rotation system: rice (Oryza sativa L. cv. Huai Dao 5), wheat (Triticum aestivum L. cv. Yangmai 39), rape (Brassica napus L. cv. Yan You 2), faba bean (Vicia faba L. cv. Suxian Can 1), hairy vetch (Vicia villosa Roth cv. Yan Tiao 4), and barley (Hordeum vulgare L. cv. Yanshi Mai 3). All varieties were certified and obtained from the Jiangsu Academy of Agricultural Sciences. For rape cultivation, a transplanting method was employed to optimize the cropping schedule and enhance establishment. Seedlings were raised be-tween September 15-20 and transplanted at approximately 35 days old, immediately following the rice harvest in mid-October. This approach resolved the temporal conflict between rape sowing requirements (September) and rice harvest timing (mid-October) while promoting robust plant development. Faba bean was dibbled.

Results

Question 8. Fig.2, 3, 4 - Give a legend (RW, RO, RV……….) below the figures.

Figure 5. The individual graphics (A, B, C, D .......) are too small and not visible well. Perhaps it would be better to present the data in a table.

Answer: Thanks very much for your suggestion. We have revised the Figure 5 citation with the detailed information for the figures and the quality of the image.

Figure 5. Effects of different rotation patterns on soil properties (0-15 cm depth). (A) soil pH in spring; (B) soil pH in autumn; (C) soil electrical conductivity (mS/cm) in spring; (D) soil electrical conductivity (mS/cm) in autumn; (E) soil available nitrogen (mg/kg) in spring; (F) soil available nitrogen (mg/kg) in autumn; (G) soil available phosphorus (mg/kg) in spring; (H) soil available phosphorus (mg/kg) in autumn; (I) soil available potassium (mg/kg) in spring; and (J) soil available potassium (mg/kg) in autumn. (Note: Treatment abbreviations: RW: Rice-wheat (control); RO: Rice-rape; RV: Rice-hairy vetch; RB: Rice-faba bean; RF: Rice-barley; RS: Rice-fallow. Different letters above bars indicate significant differences between treatments at p < 0.05. Error bars represent standard error of three biological replicates (n = 3). Soil samples were collected before rice planting in spring and after rice harvest in autumn.)

Discussion

Question 9. Lines 311-312: “Our findings largely support the initial hypothesis that incorporating diverse crops into rotation with rice would significantly improve soil fertility, reduce pest and disease pressure…….” You have no data on reduced pest and disease pressure.

Answer: Thank you for this important observation. You are correct that our study did not directly measure pest and disease pressure. And we have deleted all the pest and disease control and management in the revised manuscript.

Question 10. Lines 316-317: “As shown in Figure 2, rice following faba bean and hairy vetch exhibited increased plant height of 118.3 cm and 115.7 cm, respectively; compared to the rice-wheat control of 105.0 cm.” These values are already presented in the Results section (lines 214-215). Give the differences in relative values (%) compared to the control.

Answer: Thanks for your good suggestion to make our manuscript better. We revised the description that “Compared to the rice-wheat control of 98.44 cm, the rice-faba bean rotation resulted in a significant 4.8% reduction in plant height of 93.67 cm (p < 0.05). Other treatments also showed reduced plant heights ranging from 2.1% to 3.9% lower than the control, though these differences were not statistically significant.” on Line 354.

Question 11. Lines 320-322: “The rice-faba bean rotation achieved the highest yield of 8.73 t/ha, representing a 19.1% increase over the rice-wheat control of 7.33 t/ha. The rice-hairy vetch rotation also performed well, with a yield of 8.20 t/ha, an 11.9% increase over the control.” The same – give only relative values.

Answer: Thanks for your good comment. We revised the sentences that “All rotation treatments demonstrated yield improvements compared to the control, with the rice-faba bean rotation showing the highest increase of 18.01% over the rice-wheat control. The rice-fallow rotation also performed well, demonstrating a 15.75% yield increase. Other rotation treatments showed intermediate yield improvements ranging from 7.3% to 12.4% above the control treatment.” on Line 362.

Question 12. Lines 340-341: “The rice-faba bean rotation showed the highest K levels in spring (195 mg/kg), significantly higher than the rice-wheat control of 175 mg/kg.” This sentence from the Results section is the same as Lines 301-302 in Discussion section: In spring, the rice-faba bean rotation showed the highest K levels at 195 mg/kg, significantly higher (p<0.05) than the rice-wheat control at 175 mg/kg and all other treatments. In the discussion section, you should discuss the results in light of previous research, not repeat the results.

Answer: Thanks very much for your guidance. We revised the Lines from 400 to 414 in the revised manuscript.

The incorporation of diverse crops, especially legumes, improved overall soil nutrients, with particularly notable effects on potassium dynamics that showed distinct seasonal patterns. As shown in Figure 5I and 5J, all rotation treatments demonstrated higher K levels compared to the rice-wheat control in both spring and autumn seasons. The rice-faba bean rotation achieved the highest K levels in spring, exceeding the control by 49.68%, while the rice-rape rotation showed remarkable K enhancement in autumn, surpassing the control by 120.91%. These seasonal differences likely reflect complementary mechanisms of K cycling enhancement. The superior spring K levels under faba bean rotation can be attributed to the legume's deep root system accessing K from lower soil profiles, as demonstrated by Nuruzzaman et al. (2005). Meanwhile, the exceptional autumn K levels following rape could be attributed to its unique root exudates and residue composition that enhance K mobilization, consistent with observations by Liu et al. (2019) in similar oilseed-cereal rotations. The enhanced soil biological activity under these diverse rotations, as evidenced by our soil enzyme findings, may have facilitated more efficient K cycling throughout the year.

Reference

Liu, X., Liu, J., Xing, B., Herbert, S. J., Meng, K., Han, X., & Zhang, X. (2019). Effects of long-term continuous cropping, tillage, and fertilization on soil organic carbon and nitrogen in Chinese Mollisols. Communications in Soil Science and Plant Analysis, 50(9), 1123-1135.

Nuruzzaman, M., Lambers, H., Bolland, M. D., & Veneklaas, E. J. (2005). Phosphorus uptake by grain legumes and subsequently grown wheat at different levels of residual phosphorus fertiliser. Australian Journal of Agricultural Research, 56(10), 1041-1047.

Question 13. Lines 368-369: "For instance, Zhu et al. (2000) demonstrated that rice-wheat rotations reduced rice blast disease incidence by 27% compared to rice monoculture." = Lines 65-66: "Research in China demonstrated that rice-wheat rotations reduced the incidence of rice blast disease by 27% compared to rice monoculture (Zhu et al., 2000)." - ?????? It is not acceptable to use the same sources in the Introduction and in the Discussion.

Answer: Thanks for your good guidance. We revised the sentence on Lines 438-443.

The improved growth and yield we observed in diversified rotations likely resulted from the enhanced soil nutrient status and soil biological activity measured in our study. The higher available nitrogen, potassium and improved soil EC under legume-based rotations created more favorable conditions for rice growth. Recent work by Liu et al. (2022) found similar improvements in soil nutrient cycling and crop performance when legumes were integrated into rice-based rotations.

Reference

Liu, C., Plaza-Bonilla, D., Coulter, J. A., Zhang, S., Assefa, Y., Gan, Y., & Li, L. (2022). Diversifying crop rotations enhances agroecosystem services and resilience. Advances in Agronomy, 173, 299-335.
